# Children and young people's contributions to public involvement and engagement activities in health-related research: A scoping review

**Alison Rouncefield-Swales** [1]*, **Jane Harris** [2], **Bernie Carter** [1], **Lucy Bray** [1], **Toni Bewley** [1], **Rachael Martin** [1]

1 Faculty of Health, Social Care and Medicine, Edge Hill University, Ormskirk, United Kingdom, 2 Faculty of Health, Public Health Institute, Liverpool John Moores University, Liverpool, United Kingdom

☯ These authors contributed equally to this work.

* rouncefia@edgehill.ac.uk

## Abstract

### Background

There has been an increasing interest in how children and young people can be involved in patient and public involvement and engagement (PPIE) in health research. However, relatively little robust evidence exists about which children and young people are reported as being involved or excluded from PPIE; the methods reported as being used to involve them in PPIE; and the reasons presented for their involvement in PPIE and what happens as a result. We performed a scoping review to identify, synthesise and present what is known from the literature about patient and public involvement and engagement activities with children and young people in health related research.

### Methods

Relevant studies were identified by searches in Scopus, Medline, CINAHL, Cochrane and PsychInfo databases, and hand checking of reference lists and grey literature. An adapted version of the Guidance for Reporting Involvement of Patients and the Public (GRIPP2) was used as a framework to collate the data. Two reviewers independently screened articles and decisions were consensually made.

### Main findings

A total of 9805 references were identified (after duplicates were removed) through the literature search, of which 233 full-text articles were assessed for eligibility. Forty studies published between 2000 and 2019 were included in the review. The review reveals ambiguities in the quality of reporting of PPIE with children with clear reporting on demographics and health conditions. The review found that children and young people were commonly involved in multiple stages of research but there was also significant variation in the level at which children and young people were involved in PPIE. Evaluation of the impact of children and young people's involvement in PPIE was limited.

**Data Availability Statement:** All relevant data are within the paper and its Supporting Information files.

**Funding:** BC and LB received internal funding for dissemination and impact activities. No funding has been used to pay for researcher salaries. The funders had no role in study design, data collection and analysis, decision to publish, or preparation of the manuscript.

**Competing interests:** No authors have competing interests

## Conclusions

Consultation, engagement and participation can all offer children and young people worthwhile ways of contributing to research with the level, purpose and impact of involvement determined by the children and young people themselves. However, careful decisions need to be made to ensure that it is suited to the context, setting and focus so that the desired PPIE impacts are achieved. Improvements should be made to the evaluation and reporting of PPIE in research. This will help researchers and funders to better understand the benefits, challenges and impact of PPIE with children and young people on health research.

## Introduction

The ten year plan for patient and public involvement and engagement (PPIE) for the National Institute for Health Research in the UK was published in 2015 [1, 2] and talks about public involvement as being central to "going the extra mile" to developing high-quality research [1]. This ten year plan commits to having "a population actively involved in research to improve health and wellbeing for themselves, their family and their communities" and the "public as partners in everything we do" [1]. The plan draws upon INVOLVE's definitions of involvement being "research being carried out 'with' or 'by' members of the public rather than 'to', 'about' or 'for' them" [3], and engagement being "where information and knowledge about research is provided and disseminated" [3]. More recently, the UK standards for Public Involvement in Research [4] provides a framework for what good public involvement in research should look like and against which improvement can be assessed. The terminology in the field is subject to on-going debate with PPIE often used interchangeably, and in some cases incorrectly, with patient and public involvement (PPI). This review has taken a broad perspective and uses PPIE as an umbrella term to encompass both patient and public involvement (PPI) and patient and public engagement (PPE).

Although there is some good evidence being produced [2, 5] and many positive impacts arising from PPIE are reported, there is limited empirical 'proof' of the impact of PPIE on the quality of research [6–8]. Genuine PPIE is increasingly accepted as a necessary and fundamental component of well-designed and executed research [9, 10] that has meaningful benefits for patients [11]; yet, despite this, meaningful PPIE in research is neither consistently nor effectively utilised within all studies.

Despite pleas that the engagement of children and young people within a PPIE capacity in research studies should no longer be perceived as an optional extra [12], there is still considerable room for improvement [12]. Children and young people who participate in PPIE may benefit directly in many ways such as skills acquisition [8, 13] and gaining confidence [14]. Less direct, but nonetheless important benefits, are those accruing to the children and young people who gain meaningful benefit via guidelines and changes to clinical practice arising from research outcomes grounded in the high quality PPIE [8]. A key challenge is to ensure that the involvement of children and young people is authentic and not merely a tick box exercise [15] with tokenistic involvement operating at the bottom of Hart's ladder of participation [16].

Typically, children and young people are involved in a variety of ways in PPIE in relation to health research ranging from consultation, through to involvement, collaboration and to user-led research [15, 17]. Child and young person advisory groups [8] are one way of involving children and young people. Their input to research studies can be focused on engagement at

one time point such as the development of materials [18, 19] or involvement in data analysis [20], through to more active and continuous engagement across the lifespan of a study as seen in studies using co-design [21] and co-production [22] methodologies.

Despite there being noted benefits and challenges to PPIE, there remains inconsistency and a lack of transparency in how PPIE activities are reported which creates challenges for people wishing to review and appraise PPIE. Transparent reporting of PPIE supports future research, improves standards and best practice [23], reduces research 'waste' [24], helps increase public support and improves understanding about methods of effective involvement in research [25]. However, under-reporting and inconsistent quality of reporting of PPIE are noted in the literature [26]. The Guidance for Reporting Involvement of Patients and the Public reporting checklists (GRIPP2) [27] were developed as a means of improving the reporting of PPI. GRIPP2 aims to improve the quality, transparency, and consistency of reporting and to ensure that PPI practice is based on the best evidence. Reporting on PPIE in published research is an increasing requirement [28], but historically this was not the case and inconsistencies in the requirements of publishers mean that high-quality and consistent reporting remains low [26].

A scoping review [29] was undertaken with the aim to identify, synthesise and present what is known from the literature about patient and public involvement and engagement activities with children and young people in health related research.

Our objectives focused on:

1. which children and young people are reported as being involved or excluded in PPIE;

2. the methods reported as being used to involve them in PPIE; and

3. the reasons presented for their involvement in PPIE and what happens as a result.

## Methods

We structured the review using Arksey and O'Malley's [30] five-stage framework (identifying the research questions; identifying relevant studies; study selection; charting data; and collating, summarising, and reporting the results) for conducting a scoping review. A sixth stage was added as we also undertook a parallel 'consultation exercise' as advised by Arksey and O'Malley [30]. Additionally, we were guided by Levac et al.'s [31] recommendations that clarify and enhance the study identification, study selection, and charting the data stages of a review.

### Defining PPIE for this review

Our working definition for PPIE with children and young people for this review builds on the previously mentioned INVOLVE definitions [3] which are the most widely referred to definitions within the UK. Our intentionally broad definition of PPIE for this review–'health related research being carried out with or by children and young people rather than to, about or for them'; or, 'where information and knowledge about research is provided and disseminated' [3].

### Stage 1: Identifying the research question

An overarching research question guided our systematic search strategy and reporting of results: What is known about public involvement and engagement activities with children and young people in health related research? This question enabled us to adequately capture a broad range of existing literature while providing the opportunity for further research objectives to be added and modified throughout the review. This iterative process was useful as we became increasingly familiar with the literature.

## Stage 2: Identifying relevant studies

Search terms were developed based on consideration of the population (children and young people aged under 25 years); concept (public involvement defined as doing research 'with' or 'by' the public, rather than 'to', 'about' or 'for' the public [32] and public engagement defined as "where information and knowledge about research is provided and disseminated" [1]; and context (health research).

A comprehensive list of search terms was identified and refined through searches on Scopus and Medline by the review team. AR and JH led the development of the search strategy with the support of a health research librarian. Truncation and proximity operators were employed to increase the sensitivity of the search (see S1 Appendix for full Medline search). Searches were undertaken in Scopus, Medline, CINAHL, Cochrane and PsychInfo databases in December 2019.

The reference lists of included documents were reviewed for additional papers, and Scopus and Google Scholar were consulted to identify the citing literature. A search of the grey literature (pdf files and webpages), including a hand search, was also completed in December 2019. Grey literature was identified via Open Grey, Google and from the websites of the Royal College of Paediatrics and Child Health, Royal College of Nursing, Barnardo's, Department of Health, National Institute of Health Research Portfolio, Childlink, WellChild, Shine, Generation R, UNICEF, Save the Children, INVOLVE and iCan for research and conference publications.

## Stage 3: Study selection

Studies were included in the review if they met the criteria outlined in Table 1. The search parameters were papers published 1st January 2000 – 16th December 2019. The age range of 0-≤24 years was used to encompass children and young people and was determined based on the World Health Organization (2014) [33] definition of a 'child' as a person under the age of 18 years and a '*young person as under 24 years of age*' (10–24 years).

Citations were imported into Covidence (an online review tool that simplifies the screening and extraction process) and duplicates were removed. Following this, study selection occurred using a two-stage screening process. In Stage 1, titles and abstracts of all papers were each blind reviewed by two members of the review team (AR, JH) and conflicts resolved by a third reviewer from the team (TB, LB, BC). Seven papers, unavailable in full-text format, were

**Table 1. Inclusion and exclusion criteria.**

| **Inclusion criteria** |
| --- |
| • PPIE activities occurred with children and young people aged 0-≤24 years. |
| • General population of children, young people and young adults: but will not exclude studies which solely focus on certain chronic conditions or specific population groups as long as they meet the other inclusion criteria. |
| • Empirical and descriptive studies. |
| • Systematic reviews and meta-analysis (and those studies included in these reviews/analyses) |
| • Health related research studies (including disability research). |
| • Full text available in English. |
| **Exclusion criteria** |
| • Studies that do not include children or young people in PPIE activities. |
| • Studies focused on health service design. |
| • Book reviews, opinion pieces, unpublished theses and literature reviews. |
| • Articles in press. |

excluded. In Stage 2, all members of the review team (AR, JH, TB, LB, BC) were allocated papers with two reviewers each screening all full-text papers. Conflicts were resolved through discussion between the two reviewers or where further advice needed by a third reviewer. Where papers relating to the same study reported on different aspects of the PPIE, all relevant papers were included.

## Stage 4: Charting the data

We iteratively developed and refined a data extraction table reporting on nine key sections (Table 2) based on the Guidance for Reporting Involvement of Patients and the Public 2 (GRIPP2 short and long-form) [27]. The Guidance for Reporting Involvement of Patients and the Public–short-form and long-form (GRIPP2-SF and GRIPP2-LF) [27] reporting checklists provide key items for the reporting of PPI. The short version (GRIPP2-SF) is for reporting PPI in any study and a long version (GRIPP2-LF) is for when the study is mainly about PPI in research.

Core data were extracted on the author(s), year of publication, country of origin, research study aim, methodology, method and sample. PPIE-specific data extraction (Table 2) focused on PPIE aim, PPIE terminology, definition and underpinning concepts/theory; PPIE population; PPIE design and stages of involvement; methods by which PPIE was evaluated; evidence of the impact of PPIE; conclusions and lessons learned from PPIE; limitations related to PPIE; and recommendations arising from PPIE. Five reviewers (AR, JH, TB, LB, BC) independently extracted data from their allocated articles with extractions checked for consistency and condensed by one reviewer (BC).

**Quality assessment.**    Although assessing the methodological and other qualities of studies within a scoping review can help contextualise findings and enable interpretation, Arksey and O'Malleyhttps://onlinelibrary.wiley.com/doi/full/10.1111/hex.13069 - hex13069-bib-0019 [30] suggest that quality assessment is not required and Levac [31] recognises the difficulties in assessing quality across a wide range of published and grey literature. Despite awareness of these issues, we initially sought to report on the quality of the studies using the Mixed Methods Appraisal Tool (MMAT) [34]. However, the MMAT directed our attention towards methodological issues when we were most interested in the quality of reporting of the PPIE.

**Table 2. Overview of data extraction items based on adapted Guidance for Reporting Involvement of Patients and the Public 2 (GRIPP2) form [27].**

| Section | Item description |
|---|---|
| 1. Aim | Report the aim of PPI. |
| 2. PPIE term, definition and any underpinning concepts/theory | Report the terminology used to describe PPIE, any definition of PPIE and any underpinning concepts or theory. |
| 3. PPIE population | Describe the population involved with the PPIE activity (including age, other demographics and medical condition) and numbers involved. |
| 4. PPIE design and stages of involvement | Describe how young people were involved in the PPIE. Report the stages of research in which PPIE is used. |
| 5. Methods by which PPIE was evaluated | Outline the methods used to evaluate the impact of PPIE on the research process, on the children and young people involved and on wider policy. |
| 6. Evidence of the impact of PPIE | Describe the impact of the PPIE on the research process, on the children and young people involved and on wider policy. |
| 7. Conclusions and lessons learned from PPIE | Report the conclusions and lessons learned from PPIE. |
| 8. Limitations related to PPIE | Report the limitations of the PPIE activity. |
| 9. Recommendations arising from PPIE | Report the recommendations for future PPIE. |

No tool existed that enabled us to assess the quality of reporting from a broad range of papers, and which was not overly onerous. Having used elements of the GRIPP2 (LF- and SF) [27] to chart and extract the relevant information from the papers, we developed a short appraisal tool based on key areas of the GRIPP2-LF [27] using a similar format to the MMAT, to support our review of the quality of PPIE reporting. This tool focused on items necessary to assess the quality of the reporting for the focus of our review. One reviewer (AR) led the development of this appraisal tool. Through an iterative and condensing process involving the whole team, items from each section of the GRIPP2-LF were selected and turned into questions to develop a tool we named the QRIPPAT (Quality of Reporting Involvement of Patients and the Public Appraisal Tool). The QRIPPAT focuses on four broad themes each with several questions: PPIE aim and definition, PPIE methods, PPIE findings and discussion, and PPIE learning and reflections, all of which require only a Yes/No response (Table 3).

Assigning scores to quality appraisal can be seen as contentious, especially if the scoring system is unweighted or unvalidated. However, scoring papers can reflect a measure of their worth and the score can be used, along with other considerations, as part of the critique of a paper. With this in mind, we have equally weighted each element of the QRIPPAT and assigned a score of 1 for each Yes and 0 for each No, giving a range of 0–10, where 0 is the lowest score for quality and 10 the highest quality score. The results of the QRIPPAT assessment (Table 4) reveals the overall quality in the reporting of the PPIE in the studies under review.

## Stage 5: Collating, summarising, and reporting results

Collated and summarised information from the included papers was added to the data extraction table in S1 Table and it proved invaluable in helping to develop initial themes that were finalised through discussions involving all members of the research team. Due to the wide variation in PPIE approaches in the included studies and significant variability in the quality of reporting of PPIE, a narrative approach offered the best fit to summarise the results of the review. The NIHR's research process model [68] which outlines eight stages in the research process in which research-oriented PPIE can take place was condensed to create a better fit for reporting our findings. The findings outlined were explicitly reported by the study authors and not inferred by the authors of this review.

**Table 3. QRIPPAT (Quality of Reporting Involvement and Engagement of Patients and the Public Appraisal Tool).**

|  | Response |
| --- | --- |
| **Aim and Definition** |  |
| 1. Is there clear reporting of aim of the PPIE activity? | Yes/No |
| 2. Is there clear reporting of term used to describe PPIE? | Yes/No |
| 3. Is there clear reporting of definition of term used for PPIE? | Yes/No |
| **Methods** |  |
| 4. Is there clear reporting of PPIE population demographics? | Yes/No |
| 5. Is there clear reporting of conceptual models or influences underpinning PPIE? | Yes/No |
| 6. Is there clear reporting of methods used to engage/involve young people in PPIE? | Yes/No |
| **Findings and Discussion** |  |
| 7. Is there clear reporting of how PPIE impact has been evaluated? | Yes/No |
| 8. Is there clear reporting of impact of the PPIE? | Yes/No |
| **Learning and Reflections** |  |
| 9. Is there clear reporting of contextual / process factors that enabled/hindered PPIE? | Yes/No |
| 10. Is there clear reporting of critical learning from undertaking PPIE? | Yes/No |

**Table 4. Quality of Reporting Involvement and Engagement of Patients and the Public Appraisal Tool (QRIPPAT).**

| Author(s) (year) | Aim & Definition | | | Methods | | | Findings & Discussion | | Learning & Reflections | | Score |
|---|---|---|---|---|---|---|---|---|---|---|---|
| | Aim of PPIE activity? | Term used to describe PPIE? | Definition of term used for PPIE? | PPIE population demographics? | Conceptual models/ influences underpinning PPIE? | Methods used to engage/ involve young people in PPIE? | How PPIE impact has been evaluated? | Impact of the PPIE? | Contextual/ process factors that enabled/ hindered PPIE? | Critical learning from undertaking PPIE? | |
| Beatriz et al. (2018) [35] | Yes | Yes | No | Yes | Yes | Yes | Yes | Yes | Yes | Yes | 9 |
| Best et al. (2017) [36] | Yes | Yes | No | Yes | Yes | Yes | No | Yes | Yes | Yes | 8 |
| Boote et al. (2016) [37] | Yes | Yes | Yes | Yes | Yes | Yes | No | No | Yes | Yes | 8 |
| Brady et al. (2018) [38] | Yes | Yes | Yes | Yes | No | Yes | No | Yes | Yes | Yes | 8 |
| Byrne (2019) [39] | Yes | Yes | Yes | Yes | No | Yes | No | Yes | Yes | Yes | 8 |
| Carroll et al. (2018) [40] | Yes | Yes | No | No | No | Yes | No | Yes | No | No | 4 |
| Chopel et al. (2019) [41] | Yes | Yes | Yes | Yes | Yes | Yes | No | Yes | Yes | Yes | 9 |
| Coad (2012) [42] | Yes | Yes | No | Yes | Yes | Yes | No | No | No | Yes | 6 |
| Collin & Swist (2016) [43] | Yes | Yes | Yes | No | Yes | Yes | No | Yes | Yes | No | 7 |
| Cooper et al. (2017) [44] | No | Yes | No | No | No | Yes | No | Yes | No | No | 3 |
| Costello & Dorris (2019) [45] | Yes | Yes | Yes | Yes | Yes | Yes | Yes | Yes | Yes | Yes | 10 |
| Curtin & Murtagh (2007) [46] | Yes | Yes | No | Yes | No | Yes | No | Yes | Yes | Yes | 7 |
| Dovey-Pearce et al. (2019) [47] | Yes | Yes | Yes | Yes | No | Yes | No | No | No | Yes | 6 |
| Forsyth et al. (2019) [48] | Yes | Yes | Yes | Yes | No | Yes | No | No | No | No | 5 |
| Funk et al. (2012) [13] | Yes | Yes | Yes | Yes | No | Yes | No | Yes | Yes | Yes | 8 |
| Griffiths et al. (2018) [49] | Yes | Yes | No | Yes | No | Yes | No | Yes | No | No | 5 |
| Hannon et al. (2018) [21] | Yes | Yes | No | Yes | Yes | Yes | No | Yes | Yes | No | 7 |
| Holmes (2002) [14] | Yes | Yes | No | Yes | No | Yes | No | Yes | No | Yes | 6 |
| Hunt et al. (2013) [50] | Yes | Yes | Yes | Yes | No | Yes | No | No | Yes | Yes | 7 |
| Kendal (2017) [51] | Yes | Yes | Yes | Yes | No | Yes | No | No | Yes | Yes | 7 |
| Larkins et al. (2013) [22] | Yes | Yes | Yes | Yes | Yes | Yes | No | Yes | Yes | No | 8 |
| Liabo et al. (2018) [52] | Yes | Yes | No | Yes | No | Yes | No | Yes | Yes | Yes | 7 |

*(Continued)*

**Table 4.** (Continued)

| Author(s) (year) | Aim & Definition | | | Methods | | | Findings & Discussion | | Learning & Reflections | | Score |
|---|---|---|---|---|---|---|---|---|---|---|---|
| | Aim of PPIE activity? | Term used to describe PPIE? | Definition of term used for PPIE? | PPIE population demographics? | Conceptual models/ influences underpinning PPIE? | Methods used to engage/ involve young people in PPIE? | How PPIE impact has been evaluated? | Impact of the PPIE? | Contextual/ process factors that enabled/ hindered PPIE? | Critical learning from undertaking PPIE? | |
| Lightfoot & Sloper (2003) [53] | Yes | Yes | Yes | No | No | Yes | No | No | Yes | No | 5 |
| Locock (2019) [20] | Yes | Yes | Yes | Yes | Yes | Yes | No | Yes | No | Yes | 8 |
| Manning (2018) [54] | Yes | Yes | No | Yes | No | Yes | No | Yes | Yes | No | 6 |
| McLaughlin (2015) [55] | Yes | Yes | No | Yes | No | Yes | Yes | Yes | Yes | Yes | 8 |
| Mitchell et al. (2019) [56] | Yes | Yes | No | Yes | No | Yes | No | No | No | No | 4 |
| Mitchell et al. (2018) [57] | Yes | Yes | No | No | Yes | Yes | No | No | No | No | 4 |
| Morton et al. (2017) [58] | Yes | Yes | Yes | Yes | No | Yes | Yes | No | No | No | 6 |
| Office of Children's Commissioner (2014) [59] | Yes | Yes | No | Yes | No | Yes | No | No | No | Yes | 5 |
| O'Hara et al. (2017) [60] | Yes | Yes | Yes | Yes | No | Yes | No | Yes | Yes | Yes | 8 |
| Oliver et al. (2015) [61] | Yes | Yes | Yes | Yes | No | Yes | No | Yes | Yes | Yes | 8 |
| Pavarini (2019) [62] | Yes | Yes | Yes | Yes | Yes | Yes | Yes | Yes | Yes | Yes | 10 |
| Perry & Carpenter (2016) [63] | Yes | Yes | Yes | Yes | Yes | Yes | No | Yes | Yes | Yes | 9 |
| RCPCH (2012) [64] | Yes | Yes | No | Yes | No | Yes | No | Yes | No | No | 5 |
| Sheridan et al. (2019) [18] | Yes | Yes | Yes | Yes | Yes | Yes | Yes | Yes | Yes | Yes | 10 |
| Snodin et al. (2017) [65] | Yes | Yes | No | Yes | No | Yes | No | Yes | No | Yes | 6 |
| Taylor et al. (2015) [66] | Yes | Yes | Yes | Yes | Yes | Yes | Yes | Yes | Yes | Yes | 10 |
| Tume et al. (2016) [19] | Yes | Yes | No | Yes | No | Yes | No | Yes | Yes | Yes | 7 |
| Walsh et al. (2018) [67] | Yes | Yes | Yes | Yes | Yes | Yes | No | No | Yes | Yes | 8 |

## Stage 6: Consultation

Although not commonly used within scoping reviews, we were determined to ensure our review modelled good PPIE practice. Before commencing the review, we engaged with a young man who is an experienced service user via a service user group within our Faculty.

This discussion helped to refine the scope of the review and, through his involvement, we incorporated exploring the wider impacts of PPIE and a critical consideration of not only which children and young people are included in PPIE but also those who may be excluded. Furthermore, the review benefited from the expertise of a team member who is a skilled Family Engagement Officer (RM) who challenged and influenced the team's thinking through her insights into the priorities and areas which might be most important for children and young people involved in PPIE activities. RM provided feedback on the QRIPPAT extraction and the overall data extraction, checked the credibility of the evidence synthesis, identified gaps in the data and made suggestions as to present the findings.

Building on our earlier consultation work, the lay summary of the key points of the evidence generated from the review and recommendations for researchers is being co-produced by a group of young service users.

## Findings

In the findings, we initially report on the study characteristics, including the settings and core matters relating to research studies linked to the reported PPIE and the findings from the QRIPPAT. We then present findings relating to our core objectives:

1. which children and young people are reported as being involved or excluded in PPIE;

2. the methods reported as being used to involve them in PPIE; and

3. the reasons presented for their involvement in PPIE and what happens as a result.

### Study characteristics

The 40 articles included 38 primary research studies (the PPIE from one study was reported in three articles) [39, 60, 67]. The PRISMA diagram below (Fig 1) illustrates the article search, screening, and review process. The PPIE was undertaken as part of research conducted in the United Kingdom (n = 32) [18–20, 22, 36–39, 42, 44, 45, 47–67], Australia (n = 3) [14, 43, 46], United States (n = 3) [21, 35, 41], Canada (n = 1) [13] and New Zealand (n = 1) [40]. The majority of the articles (n = 37) were published after 2010 [13, 18–22, 35–45, 47–52, 54–67] with three published between 2000–2010 [14, 46, 53].

**Overview of study designs.**   Most articles described a single health research study although four articles [39, 46, 55, 56] described multiple health research studies within which PPIE was embedded. The most commonly reported research designs were qualitative (n = 20) [13, 18, 20–22, 36, 41–43, 45, 48, 50, 51, 53, 54, 57–59, 63, 65] and mixed-methods (n = 11) [14, 35, 39, 40, 46, 47, 55, 60, 64, 66, 67]; of these studies, five (qualitative n = 1, and mixed methods n = 4) drew on a participatory methodology [35, 36, 41–43]. The remaining studies included randomised control trials (n = 4) [19, 37, 38, 44], case study (n = 2) [49, 52] or systematic review (n = 1) [61]. The study design was unclear in two articles [56, 62].

**Quality assessment of reporting of PPIE.**   The majority of articles (n = 25) focussed almost exclusively on reporting the PPIE. Fifteen articles devoted only a small segment of the paper to reporting PPIE [14, 22, 40–44, 46, 49, 51, 57–59, 64, 67]. The diversity in the nature of the articles led to difficulties in comparing the quality of the reporting. There was a wide variation in the quality of reporting of PPIE, as is evident in our assessment using the QRIPPAT (Table 4).

Overall, there was the greater clarity in reporting on the aim of the PPIE (n = 39) [13, 14, 18–22, 35–43, 45–67], the demographics of the PPIE populations involved (n = 35) [13, 14, 18–22, 35–39, 41, 42, 45–52, 54–56, 58–67] and the methods used to engage young people in

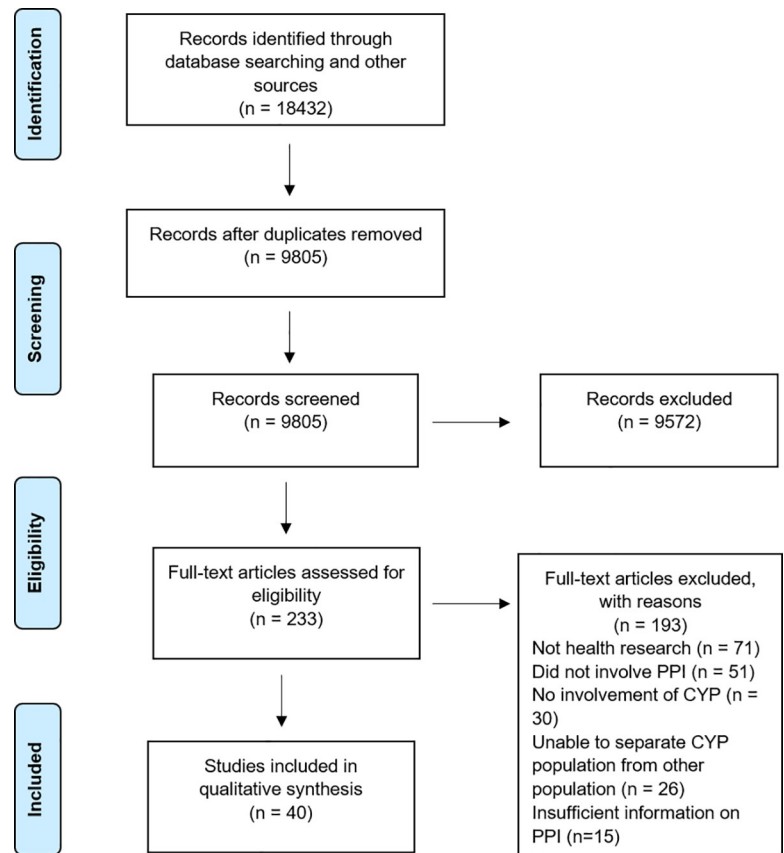

**Fig 1. PRISMA diagram.**

PPIE (n = 40) [13, 14, 18–22, 35–67]. The clarity of reporting was mixed on the PPIE definition used (n = 22) [13, 18, 20, 22, 37–39, 41, 43, 45, 47, 48, 50, 51, 53, 58, 60–63, 66, 67], the impact of PPIE (n = 28) [13, 14, 18–22, 35, 36, 38–41, 43–46, 49, 52, 54, 55, 60–66], contextual and process factors affecting PPIE (n = 26) [13, 18, 19, 21, 22, 35–39, 41, 43, 45, 46, 50–55, 60–63, 66, 67] and critical learning from the PPIE (n = 27) [13, 14, 18–20, 35–39, 41, 42, 45–47, 50–52, 55, 59–63, 65–67]. There was a lack of clarity in the quality of reporting on the conceptual influences underpinning the PPIE (n = 16) [18, 20–22, 35–37, 41–43, 45, 57, 62, 63, 66, 67] and the methods of evaluating the impact of PPIE (n = 7) [18, 35, 45, 55, 58, 62, 66].

**PPIE Terminology, definitions and conceptual and theoretical influences.**   S1 Table summarises the data extraction of Public and Patient Involvement and Engagement (PPIE) for studies included in the scoping review. A variety of terminology was reported in papers to describe the PPIE with children and young people. Commonly, the term patient and public involvement (PPI) [18–20, 36, 37, 44, 45, 47–49, 54–58, 60, 64–67] was used, alongside terms such as involvement [38, 53, 61], user involvement [50, 52], advisory [40], and engagement [39]. Where PPIE was integral to the research methodology, participatory methodology terms were frequently used included participatory design [13, 14, 43, 46, 51, 59], community-based participatory research (CBPR) [41], peer research [35], co-research [42, 63], co-design [21], and co-production [62]. One study described adopting a child rights based approach (CRBA) [22].

Explicit definition of the terminology used with reference to relevant literature was provided in 16 papers [18, 22, 37, 41, 43, 46–48, 50, 51, 53, 58, 60–62, 67]. In 14 papers [13, 19, 20, 35, 36, 38, 39, 42, 45, 54, 57, 63, 65, 66], looser definitions often without reference to literature,

were offered using terms such as research 'with' and 'by' young people. In ten papers, no definition of the terminology was provided [14, 21, 40, 44, 49, 52, 55, 57, 59, 64]. No difference in the inclusion and clarity of definition for PPIE activity was found between older or more recent publications, and the provision of a clear definition was not dependent on the terminology used.

Twenty-seven papers [13, 14, 19, 36–40, 44–55, 57, 58, 60, 61, 64–66] gave no explicit reference to any conceptual models or theoretical influences which had informed their approach to PPIE in the study. Thirteen papers referred to a conceptual or theoretical model as informing their approach to PPIE: three referred to participatory design/methodology [42, 43, 59] and the remainder to a child rights-based approach (CRBA) [22], co-operative enquiry [63], co-production model [62], empowerment theory [41], experience based co-design (EBCD) [20], patient experience framework [57], responsive and managerial public involvement [18], systems design [21], user-centred design [67] and youth participatory action research [35].

## Children and young people reported as being involved or excluded in PPIE

Most studies gave demographics of the children and young people involved in the PPIE in terms of age, with fewer (n = 22) specifying more detailed demographics, for example, gender or ethnicity, health conditions or health experiences of the children and young people [14, 18, 35, 36, 38, 40–44, 46, 48, 49, 51, 53, 56–59, 61, 62, 64]. Five studies did not report the age range of the children and young people [40, 44, 46, 53, 59]. One study specified they had engaged children younger than four years [37] and one had engaged with children and young people from 4–24 years old [22]. Seven studies had solely engaged those in young adulthood (18–25 years) [13, 18, 20, 35, 39, 60, 67] and twelve had solely engaged with adolescents (10–18 years) [21, 36, 42, 43, 45, 50, 51, 55, 58, 61–63]. No studies had engaged solely with children <11 years. Five studies had spanned childhood and adolescence (aged between 5 to 18 years) [19, 48, 54, 64, 65] and nine had spanned adolescence and young adulthood (10–25 years) [14, 38, 41, 47, 49, 52, 56, 57, 66].

Eleven studies referred to children and young people having specific conditions: arthritis [45], asthma [37], cancer [66], diabetes [21, 39, 60, 67], physical disability [22, 50], PICU experiences [19, 54], and depression or stroke [20]. Three studies addressed more general health experiences, such as the experience of accessing healthcare services [47] or a range of health experiences or conditions [55, 65]. Three studies had engaged with children and young people with other experiences that impacted health, for example, drug use [13], care leavers [52], asylum seekers [52] and experiences of therapy [63].

Typically, the numbers of children and young people involved in PPIE were small, although the range was wide from three [18] to 300 [43]. Of the 33 studies which reported the number of children and young people involved, seventeen had involved ten or fewer children and young people [13, 18, 20, 21, 35–37, 39, 41, 42, 48, 50, 59, 60, 63, 65, 67], eight had involved 11–20 children and young people [14, 19, 38, 45, 47, 51, 52, 55], six had involved 21–100 children and young people [22, 49, 54, 58, 61, 62] and two between 200–300 children and young people [43, 66].

In thirteen studies, the PPIE work involved a broader population than just children and young people and included parents [18–21, 45, 55, 64], adult stakeholders [14, 54], professionals [58] and combinations of families and professionals [22, 50, 65].

**How children and young people are recruited/identified for PPIE.** Twenty-six studies had recruited children and young people from specific populations with direct, relevant experience [13, 14, 18, 20–22, 35–37, 39–43, 45–47, 50, 52, 54, 55, 58, 60, 63, 66, 67]. Studies reported that they had recruited children and young people through schools [36], specialist

conferences [66], multimedia campaigns [39, 60] and specialist clinics or services [21, 50, 54, 60]. One study had recruited through distributing flyers and posters to groups and organisations associated with people with lived experience [20] while another outlined recruiting through existing research study stakeholders and advertising through existing networks, advisory groups and via Twitter [58]. Recruitment and retention were commonly reported as being challenging [18, 19, 21, 37, 55, 58], often resulting in the adoption of pragmatic approaches such as working with existing PPIE groups. Involvement was reported as being more meaningful and embedded rather than tokenistic when the children and young people involved in PPIE and the researchers were able to form effective working relationships [38]. Recruiting smaller subgroups of children and young people with direct experience of the topic under study was identified as essential to create a good 'fit' [55].

## Methods reported as being used to involve children and young people in PPIE

A condensed version of the NIHR research process model [68] (Fig 2) was used to guide the findings of children and young people's involvement in PPIE. PPIE activities were reported as occurring throughout various stages of a research study, and most commonly, PPIE with children and young people occurred across multiple stages. However, in twelve studies children and young people's involvement was limited to one stage of the research [19, 36, 37, 40, 44, 46, 50, 54, 61, 63, 64, 66], this was frequently within the initial stages of prioritising, design and development the research [19, 37, 44, 50, 54, 63, 64, 66]. In eight studies PPIE with children and young people occurred across two stages of a research study: prioritising, design and development, and undertaking stages [14, 48, 58, 65]; undertaking and analysis stages [42, 47]; analysis and dissemination stages [20]; and prioritising, design and development, and dissemination stages [18]. In twelve studies PPIE with children and young people occurred across three stages: prioritising, design and development, undertaking research, and analysis stages [35, 41, 43]; prioritising, design and development, undertaking research, and dissemination stages [21, 56, 57]; prioritising, design and development, analysis, and dissemination stages [67]; and, undertaking research, analysis, and dissemination stages [13, 39, 49, 51, 52]. In eight studies [22, 38, 45, 53, 55, 59, 60, 62] children and young people were involved across all four stages.

**PPIE in design and development.** Twenty-nine papers reported on children and young people's involvement in the development of research studies. Such activity was directed on establishing a focus relevant to children and young people [19, 37, 43, 45, 54, 62], defining

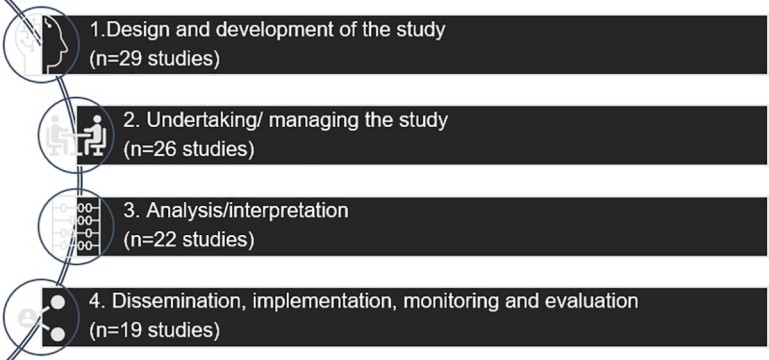

**Fig 2. PPIE involvement in the stages of the research process.**

research questions [14, 41, 50, 62, 65], reviewing protocols [48, 57], helping to establish interventions [21, 35, 38, 58], reviewing data collection methods and tools [13, 14, 18, 22, 35, 38, 40, 41, 45, 53, 55–57, 59, 60, 62–65, 67], advising on the participant recruitment strategy [44, 57, 59], advising on research population [40, 45], defining outcome measures [19, 21, 43, 44], advising on lay summaries and materials [18, 37], contributing to research ethics [19, 37, 59], and designing the study branding and logo [37, 50, 55, 56, 60, 66]. Some of this activity occurred pre-bid [37].

**PPIE in undertaking and management.** In twenty-six papers, there was evidence of children and young people contributing to the undertaking and management of the research stage. Children and young people were involved in helping to monitor and manage the research [21, 39, 46, 47, 49, 60, 65], developing participant information and materials [14, 21, 22, 38, 46, 48, 49, 52, 53, 56, 57, 59, 60, 62, 65], contributing to researcher training and support [45], designing research and publicity updates [43, 55, 62], and co-facilitating/conducting data collection [13, 14, 35, 39, 41–43, 49, 51, 55, 58].

**PPIE in analysis.** Twenty-two papers reported on children and young people's involvement in the analysis and reporting of studies. Children and young people were involved in data entry [35], developing themes and contributing to data analysis [20, 22, 35, 36, 38, 41, 42, 47, 49, 51, 52, 55, 59, 61, 62] and in supporting the interpretation of research findings [13, 39, 43, 45, 53, 60, 67].

**PPIE in dissemination.** Nineteen papers reported children and young people's involvement in PPIE in dissemination. Of these, twelve papers reported co-presenting at conferences, workshops, exhibitions and webinars [13, 38, 43, 52, 53, 55–57, 60, 62, 67]. Seven papers reported supporting the development of interventions including online campaigns [43], videos [20, 22] and health management tools [21, 39, 60, 67]. Other types of reported PPIE with children and young people included supporting further grant applications [39, 60, 67] and contributing to research papers and reports [18, 22, 38, 45, 49, 51, 56, 59, 60].

## Why PPIE is taking place and what happens as a result

Of the studies which presented evaluation methods most reported relatively simple descriptions of the impact of PPIE based on informal evaluations and often insufficient detail was presented. However, one paper acknowledged that an embedded evaluation methodology was able to offer robustness and determination of real-world meaningfulness [48].

**Methods used to evaluate the PPI.** Twenty-eight studies did not report the methods used to evaluate the impact of the PPIE interventions or strategies [13, 14, 19–21, 36, 37, 39–44, 46–48, 50–54, 57, 59, 61, 63–65, 67]. Eleven studies reported internal evaluations undertaken by the research team [18, 22, 35, 38, 45, 49, 56, 58, 60, 62, 66].

Only one study [55] formally evaluated the impact of children and young people's involvement in PPIE through an independent mixed-methods evaluation. This evaluation highlighted the success of the study and the model of participation which had substantial involvement of children and young people as co-researchers and co-producers. The young people, project personnel, parents and carers and civil servants gave positive feedback on recruitment, training and support provided to the young people finding it to be effective, well planned and positively received. Furthermore, the evaluation suggested that children and young people's life experiences shaped the study and ensured that there was a resonance with the issues under investigation.

Seven out of the eleven studies which reported internal evaluations did not report the methods used [18, 22, 35, 38, 49, 60, 66]. However, some included reflections and feedback from children and young people [18, 22, 35, 60], researchers [22, 49, 60] and post-event evaluation

[66]. Reported methods of evaluation of PPIE activity [45, 57, 58, 62] included anonymous questionnaires [62], writing workshops with children and young people and study staff [57], confidential feedback [58] and measures of effectiveness from postings on social media [45].

**Impact of involvement in PPIE on children and young people.** The reporting of the impact on children and young people was frequently anecdotal. Fourteen papers directly reported impacts on children and young people [13, 14, 20, 22, 35, 38, 41–43, 47, 55, 57, 60, 62]. Reported impacts on children and young people included gaining skills such as research and technical skills [13, 14, 35, 41, 62] and skills and confidence in public speaking [13, 62] which were viewed as improving the children and young people's chances of future employment [13, 35]. Through involvement with, and the support of professional networks [13, 60], some children and young people had found employment opportunities [13], roles within the community [41] and interest in pursuing a research career [20]. Young people were also reported as gaining softer skills such as confidence [14, 62] and advocacy skills [13, 14, 41]. Personal impacts on children and young people included empowerment and creating positive change [38, 43, 57], enjoyment [22, 42], feeling part of a team [47], feeling listened to [22], and developing a greater understanding of their rights [22]. One participatory study [41] reported a negative impact and a risk to children and young people's safety when racial discrimination was experienced during data collection. The incident resulted in improvements to the study safety protocols.

**Impact of involvement of children and young people in PPIE on the research and researchers.** Thirty-two papers reported on the impact of the involvement of children and young people in PPIE on the research and researchers [13, 14, 19–22, 35–41, 43, 44, 46, 49–52, 54, 55, 57, 59–67]. Two studies [35, 55] reflected on the considerable impact of the involvement of children and young people in PPIE throughout the whole research process which had strengthened the design, conduct and rigour of the study [35] and led to the research study having greater resonance with the issues under investigation [55]. Involvement also positively impacted on the researchers' individual understanding of PPIE processes and their skills in involving children and young people throughout the research process [36, 38, 55, 60]. For one study, this led to securing further funding and having their work recognised as having a "robust and meaningful inclusion of patient perspectives" [60].

One study reflected on the impact being "primarily about change in researchers' focus and approach to their field" [52] while another reported on the importance of involving children and young people in designing the structure and processes of the PPIE itself to ensure that timelines and expectations had a better fit with the children and young people's lives [60].

There were also clear impacts on the different stages of the research process. During the development stages, positive impacts were noted with children and young people helping to identify priorities [54], ensuring studies were informed by a lived understanding [14, 52], reducing bias [36] and improving the ethical basis of the study [37]. Studies reported the impact of children and young people in the development of patient focused outcome measures [19, 44, 63] and in enhancing the researchers' knowledge and perspective of the topic under study [52]. Four studies [49, 52, 56, 65] documented a change in the use of language due to the involvement of children and young people and in one study [56] this led to further research exploring language use in clinical settings.

During the stages of active research, children and young people provided insight into their perception of the relevance and appropriateness of the research which shaped the study focus, aims and design [19, 37, 41, 49, 50, 52, 55, 62, 65]. Involvement of children and young people had assisted in assessing the appropriateness of research tools and led to the use of more user-relevant tools [14, 39–41, 44, 46, 48, 59, 62–65]; this led to a complete change in the data collection methods in three studies [41, 62, 65]. Children and young people had assisted the design of tools by identifying lines of inquiry, helping with the wording and ensuring the acceptability

of the approach to other children and young people [14, 44, 46, 52, 59, 63, 64] as well as suggesting changes to the study population [37, 40, 59]. Children and young people helped ensure study documentation were accessible and relevant [18, 19, 37, 65] (e.g., improving the wording of patient information and invitation letters) and improved the process and wording of the consent process [19, 48]. Evidence also suggests that the involvement of children and young people assisted in improving the attractiveness of the research study to the target population by suggesting enhancements to branding [66], recruitment strategies [39, 62] and through increased credibility [14].

During the stages of analysis and reporting, children and young people helped to ensure that the identification of themes, the interpretation of data incorporated children and young people's perspectives and experiences [13, 14, 20, 22, 36, 49, 51, 52, 59, 61]. Children and young people's input ensured that children and young people perspectives were privileged in writing up [51], that key issues were raised [22] and that recommendations and implications were based on the priorities of children and young people [59, 61]. Furthermore, children and young people provided clear advice on accessible versions of documentation [59]. One study [20] reflected on improvements to engaging children and young people in data analysis and moved towards conversational engagement in the analytical process.

During the stages of dissemination and implementation, the involvement of children and young people was reported as improving dissemination due to the influence and awareness-raising in the community [14]. Furthermore, the impact of children and young people was recognised in campaigns that were reported as more inclusive and mindful of children and young people's views and experiences [21, 43].

**Impact of involvement of children and young people on policy/community.**   Several studies recognised that the impact of PPIE needs to be understood more broadly in how researchers may approach research in the future [20] and should question "how studies are positioned within a highly politicised funding-agenda" [52]. Few studies discussed the impact of the involvement of children and young people on policy with only two studies reporting a direct impact on policy [22, 41]. One study [41] led to the children and young people involved becoming vocal advocates influencing a local policy change, while the recommendations from the other study [22] were accepted by Government and local authorities. While evidence of broader impact was limited, there was an acknowledgement that papers such as those included in this review are contributing to the growing literature on the involvement of children and young people in health research [37].

## Limitations of involvement of children and young people in PPIE

Typically, the limitations of the involvement of children and young people in PPIE were reported as population issues such as recruitment and retention issues [19, 21, 37, 41, 45, 48, 50, 51] and non-representative samples/selection bias [21, 41, 45, 50, 51, 53, 54, 60, 61]. Constraints on time reduced the extent and depth that children and young people were able to be involved [13, 22, 39, 43, 51] as did resource and financial restrictions [22, 37, 43, 51, 60]. There was a need for more comprehensive training for researchers to support the involvement of children and young people [36, 52] and for the improvements to the design of activities to ensure accessibility and engagement with children and young people [55, 61]. Delays to studies can impact on young people's on-going involvement in a study and diminish the positive impact of training and capacity building for children and young people [41]. Several studies acknowledged that the robust involvement and participation of children and young people must commence at the earliest stages of a study to ensure children and young people views are integrated into planning and design [39, 46, 54, 55].

Other limitations arose relating to managing group dynamics [48], ensuring meaningful engagement and accountability [41, 43, 46, 52] and either under or over-estimation by the researchers of the children and young people's knowledge and skills [47, 52] and ability to commit to time-intensive activities [13, 52, 63]. In some studies, children and young people actively resisted the limited roles assigned to them and negotiated more active and involved roles [13, 42]. Other limitations reflect the need for researchers to be more flexible in response to young people's personal circumstances and to ensure the voices of young people "less frequently heard" are able to play an active role in the process [38]. This closely aligns with ensuring that children and young people's perspectives are not lost in translation during the research process [43, 61].

## Discussion

This scoping review has systematically examined the empirical evidence on PPIE with children and young people in health research. Our deliberately broad search terms aimed to capture the full range of PPIE activity meeting INVOLVE's definitions of involvement as research "with or by" children and young people (rather than "to, about or for them") and engagement as "where information and knowledge about research is provided and disseminated" [1, 3]. Our review found a broad spectrum of terminology to describe activities with children and young people spanning engagement and involvement through to participatory and co-production approaches. Traditionally, youth participation has been influenced by the hierarchical approach proposed in Hart's Ladder of Participation [16] with engagement and involvement of children and young people in adult initiated decisions ranked lower than approaches that provide children and young people the opportunity for shared decision-making and initiating action. However, this hierarchical approach has been criticised for failing to acknowledge that different levels of participation will be valuable in different cultural contexts [69]. In particular, these hierarchical approaches place too much emphasis on adults controlling children and young people's levels of participation, rather than acknowledging that children and young people should be able to choose how much participation they wish to have [69]. Our review found significant variation in how children and young people were being involved in PPIE with researchers emphasising the need for pragmatism and flexibility to achieve the desired PPIE impacts. Children and young people were reported as choosing to consult only on some elements of PPIE and lead others. They also selected to be less involved in some activities and more involved in others. This suggests that hierarchical approaches fail to recognise that the success of PPIE activities depend upon the preferences, abilities and availability of children and young people. Being able to judge and negotiate children and young people's preferences and abilities was facilitated by ongoing and established authentic relationships.

Our review found good quality reporting on the methods and populations involved in PPIE. However, there was mixed reporting on the definitions and impacts of PPIE and poor reporting on the underpinning concepts and evaluation of PPIE impact. This aligns with systematic reviews of PPIE reporting in health and social care research, which also found inconsistent and poor quality reporting and limited evidence on PPIE impact [26, 70]. A before and after comparison study found that these inconsistencies have persisted despite the introduction of the GRIPP reporting standards [27], suggesting uncertainty about reporting will continue until PPIE is more clearly embedded into research practice [26].

Our review found that the reporting of PPIE with children and young people was largely dependent on 1) which children and young people were involved, 2) the methods used to involve them, and 3) the impact on children and young people, the research process and on policy. We address each of these contextual factors in turn before considering how the findings

of our review can influence our reporting of effective, good quality PPIE for children and young people.

## Which children and young people are reported as being involved in PPIE

Across the studies, children and young people aged 4–24 years were involved in health related PPIE. Only seven studies [19, 22, 37, 48, 54, 64, 65] had engaged with children under ten years old. A lack of clear reporting on additional demographic factors and limited reporting on health conditions make it difficult to comment on which children and young people were excluded from PPIE involvement. However, some studies did note the underrepresentation of ethnic minority groups [53, 54] and children and young people with severe or complex needs [45, 53, 54]. Wider literature on healthcare access and transition note the additional barriers faced by children and young people from ethnic minority groups [71–73] and with chronic conditions [74, 75], suggesting PPIE is currently falling short of the ambition of involving all children and young people in research to "improve health and wellbeing for themselves, their families and their communities" [1, 2].

While the majority of studies recruited children and young people from populations with direct experience of the research topic, a substantial minority [19, 38, 44, 48, 49, 51, 53, 56, 57, 59, 61, 62, 64, 65] recruited from existing PPIE youth groups. Members of these established PPIE groups were reported as having good engagement with [38], favourable views towards health services and high health and research literacy [61]. This aligns with previous research findings which report that children and young people who are involved in PPIE tend to be more outgoing, self-confident, critical and comfortable in formal education than their non-involved peers [76, 77]. The barriers to recruiting and retaining children and young people identified in our review included constrained research timeframes [22, 37, 43], lack of funding [22, 37, 43, 60], and limits to young people's availability to attend [13, 21, 41, 51], may compound this lack of diverse representation. This suggests the limits set by research and budgetary timelines can lead to PPIE which does not fit in with the lives of children and young people, particularly if they are not involved in negotiating preferable and feasible activities and timelines in the early planning stages [77, 78].

## The methods reported as being used to involve children and young people in PPIE

The reviewed studies involved children and young people across a variety of research designs including qualitative, mixed methods, RCTs and systematic reviews. PPIE involvement took place at all phases of the research project, including development (n = 29), undertaking and management (n = 66), analysis (n = 22) and dissemination (n = 19). This breadth of involvement resonates with findings from a previous scoping review [79] and confirms that is it is feasible to effectively involve children and young people in PPIE at every stage of the research process. However, only half of the studies (n = 20) had PPIE involvement across more than two of these phases suggesting that PPIE involvement is often either not sufficiently planned and/or sustained across the life cycle of research projects.

As noted in a previous scoping review of the involvement of children and young people in research design [80], much of the PPIE involvement reported exists on the lower levels of Hart's ladder of participation [16], where young people are consulted and involved in adult initiated decisions but do not have opportunities for shared-decision making or initiating action [16]. Many of the reviewed studies noted that shorter-term, consultation based activities limited the opportunities for children and young people and researchers to build relationships [18, 39, 51] but that more meaningful and authentic engagement was achieved when young

people were given or chose to adopt more active roles [13, 40, 41, 46, 48, 62, 63]. This concurs with previous reviews which found that more active PPIE involvement can lead to more positive impacts [79, 81]. However, we would argue as do others [69], that these hierarchical views of what constitutes 'good' PPIE can create fewer opportunities for flexible PPIE based on what young people would choose, want, and have the time and assets to participate in. This is not to say that expectations of what PPIE activities can be achieved should be lowered. One study in our review described children and young people negotiating a more active PPIE role by leading workshops, data collection and analysis [42], emphasising the importance of being guided by the desire of children and young people, if evident, for greater research involvement. However, as reported by several studies in our review, moving PPIE involvement towards more active roles for children and young people is complex and requires flexibility, planning [18, 41, 47, 54, 62] and training for both children and young people and researchers [45].

## Why PPIE is taking place and what happens as a result

The impact of PPIE is a key aspect of the UK standards [4] and encourages researchers to identify and share the benefits, changes and learning gained from public insights. Despite increased commitment among researchers, funders and policymakers to demonstrate the positive impacts of PPIE involvement [1, 2, 82], the evidence base on the impacts of PPIE remains patchy and largely observational [80, 83]. It has been suggested that difficulties in assessing the impact of PPIE arise due to the complexity of PPIE [84] and that impact is highly contingent on context and the precise nature of involvement [85].

In agreement with a previous systematic review on the impacts of adult PPIE [81], our review suggests a wide range of positive PPIE impacts for children and young people, research, policy and the wider community across all phases of the research process. Evidence from the reviewed studies suggests that the involvement of children and young people help to refine research priorities, increase the accessibility and attractiveness of research methods and ensure children and young people's perspectives are represented in analysis, outputs and dissemination. A range of positive individual (for example, new research skills or increased confidence) and interpersonal (for example, professional and community networks) impacts were reported for children and young people. Two studies suggested longer-term impacts including employment [35] and community advocacy roles [13]. However, in line with previous research [80, 83], the majority of this evidence on impacts was anecdotal, with only seven studies using formal methods to evaluate and evidence the impact of PPIE. While the GRIPP-2 reporting standards have the potential to improve the reporting of PPIE impact, this evidence base can only be developed through the inclusion of formal, planned evaluation activities during the early research development stages [83]. However, we would again argue against the hierarchical view of evaluation that prioritises statistical evidence of impact [86]. The diverse range of impacts noted in the reviewed studies suggests a context specific approach to impact evaluation should be taken [86] which consults with children and young people, to ensure that PPIE has positive impacts not only for researchers and research funders but also for the children and young people who have volunteered their time to participate [82].

Consistent with the search strategy which was focused on health research rather than health policy, only two studies had reported positive impacts of PPIE with children and young people on policy which is unsurprising as policy impacts tend to take longer to be realised and often fall outside of the timescales of a research project. The numerous life transitions experienced by children and young people such as moving schools, colleges and universities; transitioning from child to adult services; and age restrictions on organised groups can make it more difficult for researchers to sustain relationships and measure long term outcomes once their

research has been concluded. Research funders should therefore consider how resources allocated for PPIE can be most effectively used, including allowing continued activity and follow-up beyond the time-restrictions of the research project.

## Implications for practice

Our scoping review found that PPIE with children and young people is feasible across all phases of research for a range of ages, populations and contexts. Efforts should be made to prevent the exclusion of children and young people from minority ethnic groups, younger children, those with severe or complex need, lower socio-economic groups and other marginalised populations from PPIE. Smaller groups, varied opportunities for participation according to abilities and preferences, and community organisation partnership can allow more diverse groups of children and young people to be recruited and retained. Engagement with children and young people appears more meaningful and authentic when it moves beyond consultation to allow them opportunities for active involvement and shared decision making across the life course of the project.

PPIE is widely regarded as having positive impacts on children and young people and research, but evaluation of PPIE activity must be incorporated at the planning stage to ensure these impacts are evidenced.

Overall, our review found that reporting of PPIE activity with children and young people was variable and there was poor quality reporting of definitions, underpinning theory, and the evaluation of PPIE impact. Despite the existence of the GRIPP-2 quality reporting standards [80], uncertainty is evident around reporting of PPIE for children and young people, particularly as it is often not a requirement of research publications.

We would encourage researchers and practitioners to embrace a broader understanding of what effective, good quality PPIE can be in the context of their work with children and young people. Based on the findings of this scoping review we recommend a new, non-hierarchical perspective on PPIE for children and young people that encompasses, supports and values all elements of participation by acknowledging that a flexible, context specific approach to involvement and engagement, led by children and young people's choice, is necessary.

PPIE with children and young people should be pragmatic, flexible, and suited to the context, setting and the focus of the study. It should be led by children and young people's preferences, choices, abilities and interests and should respect their time, skills, and commitment. Consultation, engagement and participation can all offer children and young people worthwhile ways of contributing to research with the level, purpose and impact of involvement determined by the children and young people themselves.

## Strengths and limitations of the scoping review

Although we used a wide range of search terms to capture the variation in PPIE terminology, inconsistencies in the keywords and terms used for PPIE means some studies may not have been included and this may have limited the findings. The lack of formal evaluation and the lack of standardised and clear reporting of PPIE led to difficulties in extracting relevant information from the papers and limits what can be reported in our review. Few journals require information about PPIE, which may lead to the under-reporting of PPIE activities. While the British Medical Journal (BMJ) has introduced the requirement for specific information about PPI, a recent study found that only 11% of studies published in the BMJ reported PPI activity [26].

Despite these limitations, our study has strengths such as our commitment to PPIE. Our search was strengthened by the involvement of a PPIE expert (RM) and a service user who

helped us define our research questions, focus our data extraction and this input strengthened our findings. A further strength of our study was our commitment to overcome the lack of a suitable tool for appraising the quality of PPIE reporting. To address this, we have developed a simple tool for quality appraisal of PPIE research. This tool, the QRIPPAT (Quality of Reporting Involvement of Patients and the Public Appraisal Tool), was developed from a validated reporting checklist, the GRIPP2 [27], and has potential to be developed further and used to evaluate ca the quality of PPIE reporting across multiple studies.

## Conclusion

Our review notes that PPIE extends beyond the boundaries of INVOLVE's accepted definition of "research being carried out 'with' or 'by' members of the public rather than 'to', 'about' or 'for' them" and engagement being "where information and knowledge about research is provided and disseminated"[3]. PPIE is more complex and involved than this simple definition suggests and allows. Children and young people need researchers to adopt pragmatic and flexible ways of working and their preferences, choices, abilities and interests need to be central to the way that researchers work with them. Consultation, engagement and participation can all offer children and young people worthwhile ways of contributing to research with the level, purpose and impact of involvement determined by the children and young people themselves. However, careful decisions need to be made to ensure that it is suited to the context, setting and focus so that the desired PPIE impacts are achieved.

Improvements should be made to the evaluation and reporting of PPIE in research. This will help researchers and funders to better understand the benefits, challenges and impact of PPIE with children and young people on health research. Our review further concludes that PPIE should be consistently and clearly reported in all studies and with further development the QRIPPAT could offer a clear way forward for quality appraisal of PPIE research.

## Implications

PPIE with children and young people should be pragmatic, flexible and led by children and young people's preferences, choices, abilities and interests and should respect their time, skills, and commitment.

## Supporting information

**S1 Appendix. MEDLINE search strategy.**
(PDF)

**S1 Table. Full data extraction table.**
(PDF)

## Author Contributions

**Conceptualization:** Alison Rouncefield-Swales, Jane Harris, Bernie Carter, Lucy Bray, Toni Bewley.

**Data curation:** Alison Rouncefield-Swales, Jane Harris.

**Formal analysis:** Alison Rouncefield-Swales, Jane Harris, Bernie Carter, Lucy Bray, Toni Bewley, Rachael Martin.

**Funding acquisition:** Bernie Carter.

**Investigation:** Alison Rouncefield-Swales, Jane Harris, Bernie Carter, Lucy Bray, Toni Bewley, Rachael Martin.

**Methodology:** Alison Rouncefield-Swales, Jane Harris, Bernie Carter.

**Project administration:** Alison Rouncefield-Swales.

**Supervision:** Bernie Carter.

**Validation:** Alison Rouncefield-Swales, Jane Harris, Bernie Carter, Lucy Bray, Toni Bewley, Rachael Martin.

**Writing – original draft:** Alison Rouncefield-Swales, Jane Harris.

**Writing – review & editing:** Bernie Carter, Lucy Bray, Toni Bewley.

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
