## [Decision Letter · Decision Letter 0]

8 Feb 2021

PONE-D-20-30639

Children and young people’s contributions to public involvement and engagement activities in health-related research: a scoping review.

PLOS ONE

Dear Dr. Alison Rouncefield-Swales,

Thank you for submitting your manuscript to PLOS ONE. After careful consideration, we feel that it has merit but does not fully meet PLOS ONE’s publication criteria as it currently stands. Therefore, we invite you to submit a revised version of the manuscript that addresses the points raised during the review process.

This is an important manuscript to widen involvement of children and young people in research. Please respond to the peer reviewer comments detailed below. Please respond to all the comments received  to strengthen the quality of your reporting, in particularly ensure you have clearly defined Patient and Public Involvement in the context of your review. This is important for readers less familiar with the concept of PPI and why this is important for research. Please ensure to carefully proof read your revised submission. 

We look forward to receiving your revised manuscript.

Kind regards,

Catherine J Evans, PhD, MSc, BSc (Hons)

Academic Editor

PLOS ONE

Journal Requirements:

Reviewers' comments:

Reviewer's Responses to Questions

**Comments to the Author**

1. Is the manuscript technically sound, and do the data support the conclusions?

Reviewer #1: Yes

Reviewer #2: Yes

Reviewer #3: Partly

Reviewer #4: Partly

2. Has the statistical analysis been performed appropriately and rigorously? 

Reviewer #1: N/A

Reviewer #2: N/A

Reviewer #3: N/A

Reviewer #4: N/A

3. Have the authors made all data underlying the findings in their manuscript fully available?

Reviewer #1: Yes

Reviewer #2: Yes

Reviewer #3: Yes

Reviewer #4: Yes

4. Is the manuscript presented in an intelligible fashion and written in standard English?

Reviewer #1: Yes

Reviewer #2: Yes

Reviewer #3: Yes

Reviewer #4: Yes

5. Review Comments to the Author

Reviewer #1: Thanks for the opportunity to review this article, which is a well written scoping review of children and young people's PPIE activities in health related research. It addresses an important issue and adds to the growing literature in this area. There are a few suggestions that would enhance the article:

The number of authors/reviewers is confusing. There are 6 listed, but the text refers to five? It is also incorrect to state that the authors contributed equally to the work, when only two authors undertook the initial screening process.

In the introduction, a reference to CYP's tokenistic involvement in research and Hart's ladder of participation would be helpful, particularly as this is appropriately critiqued later on.

The aims of the review should be at the end of the introduction rather than at the beginning of the methods.

In the study selection, what is the justification for limiting the search to papers published after 1 January 2000?

Inclusion criteria - does including systematic reviews present the possibility of double reporting? Would it not be better to search the included studies of relevant systematic reviews?

Exclusion criteria - I don't understand what is meant by 'Articles in press'.

On page 8, the phrase 'a good fit' feels unscientific - did papers meet the inclusion criteria or not?

The development of the QRIPPAT tool is to be applauded. However Table 4 would benefit from the papers being referenced and a score of the quality of each paper would enable comparison.

The paper would benefit from a proof read, there are some typos.

On the whole, it is a great paper and like all good papers, has forced me to reconsider my own practice in this area. Well done.

Reviewer #2: This is a timely and exhaustively prepared piece of work with a solid method. It follows all the right current trends (e.g. GRIPP2), it is well-read (e.g the use of Levac was excellent) and is a well-intentioned piece of work. This is important and gives INVOLVE much to think about whilst reviewing its scope. It also enables researchers to see with clear evidence what works and works well for PPIE. Good article which needs disseminating widely.

Reviewer #3: Many thanks for the opportunity to review this article.

It provided interesting results however I had some concerns over the distinction between Patient and Public Involvement (PPI) and Public Engagement (PE). The difference between "involvement" and "engagement" is briefly highlighted but I would disagree that it’s now a broad term used by many health organisations. The use of these terms is a continuing debate in this area and many consider people to use the terms incorrectly often referring to public engagement as PPI or labelling PPI as public engagement, when they are very different activities. It would be better for the authors’ say they are choosing to use the term PPIE throughout the article to encompass both PPI and PE activities.

In addition, it felt like some of the current debates in this area weren’t fully explored or acknowledged: how impact should be measured, payment for PPI/PE. There was also no mention of the NIHR UK Standards for Public Involvement in Research and how any of these studies met (or didn’t meet) these.

The articles would benefit from being carefully checked for spelling grammar mistakes as there were a few.

Below are more specific comments.

Introduction:

Line 61: PPI is widely accepted as referring to “Patient and Public Involvement” rather than “Public and Patient Involvement”

Line 61: “The term PPI rather than for public and patient involvement…” is there a word missing as this sentence isn’t clear.

Line 66: while there is evidence that meaningful PPIE is an essential part of research, there is (unfortunately) still a reluctance by many to include it in their research

Line 69: unfortunately many researchers/clinicians do see PPIE as an optional extra. It is not compulsory on all funders’ application forms. Even with those that do include PPIE requirements, there is a gap between what is include on application forms and what happens in practice.

Methods:

It would be interesting to include how many of the ways for measuring the impact of PPIE view PPIE as something that can be evaluated, for example and intervention

Line 99: “PPI” is used here rather than “PPIE”?

Line 111: The definition given is the most widely known definition of PPI in the UK, but it is not PPIE. Engagement is not “research carried out with or by children and young people”. Engagement, as defined in the introduction, is “where information and knowledge about research is provided and disseminated”.

Line 123: why was 25 chosen as the age limit for “young people”

Line 123: here public involvement is defined correctly but public engagement is missed out?

Line 126: authors’ initials missing “XX”

Line 139: why was 1 January 2000 chosen as the earliest date when searching?

Line 147, 148, 149, 167, 183: authors’ initials missing “XX”

Line 159: The GRIPP tool is for reporting PPI not public engagement

Line 206: How did the scope change after involvement from the service user?

Line 209: Family Engagement Officer’s initials missing

Line 209: It would be interesting to hear more about changes made after challenges from the Family Engagement Officer.

Line 213: “Servicer user” is without a hyphen in line 205 but with a hyphen in line 213. It would be better if these could be the same for consistency

Findings:

Line 219: the “are” is not included in the objectives in line 100

Line 324: Line 404 – “Were” is in capitals and doesn’t need to be

Line 408: Good to include the negative impact as well as positives. Did the study report how they support the CYP through this?

Line 411: the word “people” is missing from the title

Line 473: Should this be PPI or PPIE here?

Discussion:

It would be interesting to see more information included on why exactly there is difficulty in evaluating the impact of PPIE in research

It’s unusual to have some of the headings in the discussion section posed as questions and that the headings are the same as the ones used in the findings section.

Reviewer #4: The authors present a well-conducted scoping review on patient and public involvement and engagement (PPIE) with children and young people in health-related research. Their findings highlight some important areas for improved practice around reporting characteristics of PPIE members, involving people throughout the research cycle, and more rigorously capturing impact. Below are some suggestions intended to strengthen the manuscript.

Abstract:

1) There is currently some blurring between the methods/findings (e.g. number of studies identified is in the methods) and findings/conclusions (e.g. there are authors reflections under findings).

2) Please consider being more specific in the findings and conclusions. For example, it’s currently it’s not clear what was variable about their involvement, and the conclusions feel quite general (rather than linked to the specifics of your interesting review findings).

3) It might be more meaningful to present the number of records after duplicates were removed (n=9805) to reflect the number screened.

Introduction

4) I’m not sure I’d agree that PPIE is now used instead of PPI – it depends on if people are conducting ‘engagement’ as well as ‘involvement’ activities. The NIHR defines these as quite different activities: https://www.invo.org.uk/public-involvementparticipationengagement-in-research/. It’s fine if this review includes both engagement and involvement, but the definition in the methods only covers involvement. Please could this be clarified?

5) I was puzzled by the focus on reporting in the introduction as this does not relate to the review objectives – please consider whether some of this may be better suited to the discussion.

Methods

6) Please could the authors clarify the types of grey literature which were used? Searching for grey literature is described, but then the exclusion criteria excludes unpublished theses and articles in press.

7) Please consider holding back on findings (e.g. the number of papers screened / included, the reporting quality assessment) until the results section.

8) Please can the authors clarify why the GRIPP-2-LF was not sufficient to assess reporting quality?

9) Page 12 line 192 – Please consider rephasing as this currently feels as though it conflates reporting quality with the quality of the work.

Results

10) The results section currently includes a lot of information, and it’s not always clear that these relate to the review objectives – e.g. the sections on PPIE terminology, definitions and theoretical influences; methods to evaluate PPI, limitations of involvement. Please consider whether these are necessary to include, as they may distract from the core review findings.

11) Table 4: This information is well-summarised in the text, I wonder if the table itself would be better as a supplement (and potentially swapping this for the PRISMA flow chart, which would be helpful in main text).

12) Page 20 lines 297 to 304 lists how many were involved in each study. As the results section is already quite long, this might not be necessary to include in the main text.

13) Page 20 lines 311 notes that 28 studies recruited people with specific experience. As the objective for this section is ‘how’ people are recruited, please could more detail be provided? Readers may find this help to provide ideas of how they can reach out to PPIE members themselves.

14) There are a few things in the results that may be better placed in the methods. For example, page 20 lines 320-324 on how the NIHR research process model was used to categorise findings, and page 24 line 394 stating that impacts were taken directly from authors reporting and not inferred.

Discussion

15) The first couple of sentences of the introduction might not be needed as they repeat aspects of the introduction/aim. It was also unclear how bringing in the comments on the hierarchical approach related to the findings.

16) I worry that although the point about reporting is important, this was not part of the review objectives, and the amount of focus on this aspect is drawing away from the other important findings (e.g. about gaps in involvement across the research cycle, about the need for more rigorous evaluation activities).

17) I’d be hesitant to highlight that only two studies reported impact on policy, as the search strategy for the review was particularly focused on PPIE in the context of health research (rather than searching for reports of involvement in health research and policy, for example).

18) As the authors have done well in the earlier discussion paragraphs so make the implications quite clear, the 'implications' paragraph feels like it contains some repetition.

19) I would suggest recommending further work to develop the QRIPPAT for potential use to assess reporting quality, rather than recommending use in it’s current form. It’s important to note that the EQUATOR group recommend reporting guidelines are developed using a formal process, including a consensus exercise: https://journals.plos.org/plosmedicine/article?id=10.1371/journal.pmed.1000217#s1

20) Similar to the abstract, please revise the conclusions to more specifically summarise the review findings. In line with above, I’d remove the recommendation of using the QRIPPAT as this was not the focus of this project, and it was not developed with an explicit methodology.

6. PLOS authors have the option to publish the peer review history of their article (what does this mean?). If published, this will include your full peer review and any attached files.

Reviewer #1: No

Reviewer #2: No

Reviewer #3: No

Reviewer #4: No

---

## [Author Response · Author response to Decision Letter 0]

2 Mar 2021

Dear Catherine J Evans,

First, we would like to thank all the reviewers for their careful reading of the manuscript and their constructive remarks. We were impressed by the thoroughness of the reviews and appreciate the time and thoughtfulness that all four reviewers have put into the feedback. We have provided a detailed list of our responses to the reviewers’ comments in the document 'Response to Reviewers' and note the changes we have made to the manuscript. 

We hope the reviewers agree that by addressing their comments we have produced a much stronger paper. 

Many thanks again for considering our paper for publication.

Best wishes,

Alison Rouncefield-Swales (corresponding author; writing on behalf of all authors).

---

## [Decision Letter · Decision Letter 1]

7 May 2021

PONE-D-20-30639R1

Children and young people’s contributions to public involvement and engagement activities in health-related research: a scoping review.

PLOS ONE

Dear Dr. Rouncefield-Swales, 

Thank you for submitting your manuscript to PLOS ONE. After careful consideration, we feel that it has merit but does not fully meet PLOS ONE’s publication criteria as it currently stands. Therefore, we invite you to submit a revised version of the manuscript that addresses the points raised during the review process.

ACADEMIC EDITOR: 

This is a robustly conducted and carefully considered scoping review that makes an important contribution to the involvement and engagement of children and young people in health service research. You have responded to the peer review comments thoroughly. This has strengthened the quality and clarity of your reporting.

I have a few minor comments regarding presentation. Please review and revise:

1. Table 4- reporting the quality assessment. This would be better placed in the results to follow section on Quality Appraisal for the reader to review your narrative reporting and table 4. The Quality appraisal section should also come after the section 'Overview of stuy designs' . The reader can then consider your findings in the context of the quality of the included studies. In the Quality Appraisal section cite table 4, which is reporting your results, not table 3 which is reporting your methods.

2. Supporting information - please ensure the Supporting Information is referred to in the manuscript. I can see SI Full Medline search line 138. But not Supplementary Information Table Reporting full data extraction. Can you detail this in the Results section 'Overivew of the study designs' to inform the reader that supplementary information overviews the included studies. The sub-heading should state Designs - plural. The supplementary table is called Table 3 - this is incorrect, can you update.

3. Figure 1 - PRISMA Flow Diagram is listed as Supplementary Information. But, is presented in the manuscript. Can you delete from the SI list, to be clear that this figure is included in the main manuscript. Also figure 2 is listed as Supplementary Information. Can you delete from Supplementary Information, to be clear the figure is to be included in the main manuscript

4. Reviewer #4 (see below) please respond to typo - **Only minor thing - it looks like there might be a typo/tracked changes error in the conclusion, as there is a sentence that repeats: "Improvements should be made to the evaluation and reporting of PPIE in research" - appears at page 36 line 702 and page 37 line 705.**

We look forward to receiving your revised manuscript.

Kind regards,

Catherine J Evans, PhD, MSc, BSc (Hons)

Academic Editor

PLOS ONE

Journal Requirements:

Reviewers' comments:

Reviewer's Responses to Questions

**Comments to the Author**

1. If the authors have adequately addressed your comments raised in a previous round of review and you feel that this manuscript is now acceptable for publication, you may indicate that here to bypass the “Comments to the Author” section, enter your conflict of interest statement in the “Confidential to Editor” section, and submit your "Accept" recommendation.

Reviewer #4: All comments have been addressed

2. Is the manuscript technically sound, and do the data support the conclusions?

Reviewer #4: Yes

3. Has the statistical analysis been performed appropriately and rigorously? 

Reviewer #4: N/A

4. Have the authors made all data underlying the findings in their manuscript fully available?

Reviewer #4: Yes

5. Is the manuscript presented in an intelligible fashion and written in standard English?

Reviewer #4: Yes

6. Review Comments to the Author

Reviewer #4: Thank you to the authors for their clear and detailed responses and edits, which have answered all my queries. Looking forward to seeing this published, it's a really useful review.

**Only minor thing - it looks like there might be a typo/tracked changes error in the conclusion, as there is a sentence that repeats: "Improvements should be made to the evaluation and reporting of PPIE in research" - appears at page 36 line 702 and page 37 line 705.**

7. PLOS authors have the option to publish the peer review history of their article (what does this mean?). If published, this will include your full peer review and any attached files.

Reviewer #4: No

---

## [Author Response · Author response to Decision Letter 1]

12 May 2021

Article: Children and young people’s contributions to public involvement and engagement activities in health related research: a scoping review.

Dear Catherine J Evans,

First, we would like to thank you and the reviewers for their careful reading of the revised manuscript. We have responded to the four comments and have made the required edits to manuscript. 

Many thanks again for considering our paper for publication.

Best wishes,

Alison Rouncefield-Swales (corresponding author; writing on behalf of all authors). 

Our response to the reviewer and editor comments are highlighted in the Response to Reviewers document and in order below:

1. We have moved Table 4 to the results section and have moved the ‘Quality assessment’ section to after the ‘Overview of study designs’ and have changed the reference to Table 3 to Table 4.

2. Thank you for drawing attention to this error. We have changed the title of the supplementary table to S1 Table. We have referenced the inclusion of the material in the results section just after the sub-heading for PPIE (Line 277) as the extraction table refers solely to PPIE activity.

3. Thank you we have deleted reference to Fig 1 and 2 from the Supplementary Information.

4. Thank you for picking up this typo. We have deleted lines 703-705.

---

## [Editor Report · Decision Letter 2]

24 May 2021

Children and young people’s contributions to public involvement and engagement activities in health-related research: a scoping review.

PONE-D-20-30639R2

Dear Dr. Rouncefield-Swales,

We’re pleased to inform you that your manuscript has been judged scientifically suitable for publication and will be formally accepted for publication once it meets all outstanding technical requirements.

Kind regards,

Catherine J Evans, PhD, MSc, BSc (Hons)

Academic Editor

PLOS ONE
---

## [Editor Report · Acceptance letter]

28 May 2021

PONE-D-20-30639R2 

Children and young people’s contributions to public involvement and engagement activities in health-related research: a scoping review. 

Dear Dr. Rouncefield-Swales:

I'm pleased to inform you that your manuscript has been deemed suitable for publication in PLOS ONE. Congratulations! Your manuscript is now with our production department. 

Kind regards, 

on behalf of

Dr. PLOS Manuscript Reassignment 

Staff Editor

PLOS ONE